# Predicting adherence to postdischarge malaria chemoprevention in Malawian pre-school children: A prognostic multivariable analysis

Melf-Jakob Kühl[1,2]*, Thandile Nkosi-Gondwe[3,4], Feiko O ter Kuile[5], Kamija S Phiri[3,4], Mehmajeet Pannu[1], Mavuto Mukaka[6,7], Bjarne Robberstad[2], Ingunn M. S Engebretsen[1]

1 Department of Global Public Health and Primary Care, Centre for International Health (CIH), University of Bergen, Bergen, Norway, 2 Department of Global Public Health and Primary Care, Health Economics Leadership and Translational Ethics Research Group, University of Bergen, Bergen, Norway, 3 School of Global and Public Health, Kamuzu University of Health Sciences, Blantyre, Malawi, 4 Training and Research Unit of Excellence (TRUE), Blantyre, Malawi, 5 Department of Clinical Sciences, Liverpool School of Tropical Medicine (LSTM), Liverpool, United Kingdom, 6 Mahidol-Oxford Tropical Medicine Research Unit, Mahidol University, Bangkok, Thailand, 7 Nuffield Department of Medicine, Centre for Tropical Medicine, University of Oxford, Oxford, United Kingdom

* melf-jakob.kuhl@uib.no

**Data Availability Statement:** We use data collected by our consortium during a trial in Malawi. The results and all data are available via the

## Abstract

Chemoprevention with antimalarials is a key strategy for malaria control in sub-Saharan Africa. Three months of postdischarge malaria chemoprevention (PDMC) reduces malaria-related mortality and morbidity in pre-school children recently discharged from hospital following recovery from severe anemia. Research on adherence to preventive antimalarials in children is scarce. We aimed to investigate the predictors for caregivers' adherence to three courses of monthly PDMC in Malawi. We used data from a cluster randomized implementation trial of PDMC in Malawi (n = 357). Modified Poisson regression for clustered data was used to obtain relative risks of predictors for full adherence to PDMC. We did not find a conclusive set of predictors for PDMC adherence. The distribution of households across a socio-economic index and caregivers' education showed mixed associations with poor adherence. Caregivers of children with four or more malaria infections in the past year were associated with reduced adherence. With these results, we cannot confirm the associations established in the literature for caregiver adherence to artemisinin-based combination therapies (ACTs). PDMC combines multiple factors that complicate adherence. Our results may indicate that prevention interventions introduce a distinct complexity to ACT adherence behavior. Until we better understand this relationship, PDMC programs should ensure high program fidelity to sustain adherence by caregivers during implementation.

trial results' publication: Nkosi-Gondwe T, Robberstad B, Mukaka MI, R., Opoka R, Banda S, Kühl M-J, et al. Adherence to community versus facility-based delivery of monthly malaria chemoprevention with dihydroartemisinin-piperaquine for the post-discharge management of severe anemia in Malawian children: A cluster randomized trial. PLOS ONE. 2021;16(9): e0255769. doi: 10.1371/journal.pone.0255769.

**Funding:** The study was funded by the Research Council of Norway through the Global Health and Vaccination (GLOBVAC) Programme (project number 234487), which is part of the European and Developing Countries Clinical Trials Partnership (EDCTP2), supported by the European Union. The funders played no role in the study's design, data collection, analysis, write-up, or the decision to submit it for publication.

**Competing interests:** The authors have declared that no competing interests exist.

## Introduction

Malaria-related anemia has caused high mortality and morbidity and remains a leading burden of disease in the child population in Malawi, especially in highly endemic areas [1–4]. A recent meta-analysis estimated that for sub-Saharan Africa, the odds of dying among children during the first six months after their treatment for severe anemia are 72% higher than during the treatment phase in hospital, and over two times higher than for those admitted with other conditions [5]. In June 2022, the World Health Organization (WHO) recommended post-discharge malaria chemoprevention ('PDMC', previously called 'PMC' and 'IPTpd') in the updated malaria chemoprevention guidelines for settings with moderate to high malaria transmission [6]. PDMC comprises three months of malaria chemoprevention provided as monthly treatment courses with long-acting antimalarials to preschool children recently discharged from hospital after recovery from severe anemia. A recent multi-center randomized controlled trial (RCT) in Uganda and Kenya provided three months of PDMC with monthly dihydroarte-misinin-piperaquine (DP) and found a 70% protective effect against readmission and death during the intervention period [7]. A cluster randomized implementation trial in Malawi assessed adherence to PDMC following different distribution methods of the same PDMC regimen [8]. Full adherence by caregivers who received all three courses of DP at discharge (community-based PDMC) was 44% higher than adherence to a monthly regimen requiring the collection of each course at the hospital (facility-based PDMC). While the main finding of community-based PDMC yielding higher adherence was clear, key underlying determinants influencing adherence to PDMC, beyond the delivery strategy, remain poorly understood.

Evidence suggests relatively poorer overall adherence to antimalarial therapy in infected young children, cared for by their caregivers, than adherence in adults with malaria [9, 10]. Among caregivers, older age, higher education, literacy, and perception of disease severity have been associated with better adherence to their children's therapy [11, 12]. However, predictors for caregiver adherence to malaria treatment in sick children may not apply when using the same drugs for chemoprevention. While chemoprophylactic antimalarial use in infants (perennial malaria chemoprevention (PMC), previously IPTi) and school children (IPTsc) has been more researched, these strategies are delivered in line with established immunization platforms or school schedules [13–15]. Adherence predictors for these interventions are, therefore, not directly applicable to PDMC either. Using data from the implementation trial in Malawi, we thus developed a prognostic multivariable model to investigate potential determinants of PDMC adherence among caregivers from mainly rural communities. We aim to inform national malaria programs in sub-Saharan countries with moderate to high malaria transmission that plan to implement PDMC.

## Materials and methods

### Design and participants

This study is a secondary analysis of data collected in the PDMC delivery mechanism trial conducted in Malawi, described elsewhere in detail [16]. In short, the cluster-randomized controlled trial assessed two PDMC distribution strategies of the monthly DP regimen in children discharged from hospital after recovery from severe anemia. Children were randomized to receive PDMC using either a community or a facility-based distribution scheme. In addition, two reminder mechanisms (use of short text messages or community health worker reminders) were factorially added to the distribution strategies [8]. However, we disregarded them in this analysis as they did not significantly affect adherence.

We included data from 357 children who were accompanied by their main caregivers and completed the study (Fig 1). Sample size calculations and management of missing data have been published alongside the trial results [8]. Between March 2016 and July 2018, children aged <5 years, living within Zomba District in Southern Malawi whose caregivers gave informed consent were enrolled upon discharge from Zomba Central Hospital. The 3-months follow-up period ended in October 2018. Children not accompanied by their main caregiver were excluded because reliable information on the child and household could not be obtained. The district's 1460 villages (clusters) were randomly allocated to either PDMC delivery arm. Participants from the same village received the same PDMC distribution strategy. Participants in the community-based distribution arm were given the full regimen of 9 tablets upon hospital discharge and instructed to administer it as three monthly courses of a once-daily tablet for three days, starting two weeks after discharge. Participants in the facility-based arm received the same regimen. However, they had to collect the PDMC courses at prescribed monthly intervals from the hospital pharmacy [16]. Both delivery strategies required caregivers to remember to give the medication at the correct intervals or to collect subsequent treatment courses and administer them as instructed.

## Ethics statement

This study is part of the PDMC trial in Malawi. It received ethical approval from the research ethics committees of the College of Medicine in Malawi (COMREC, approval number P·02/15/1679) and the Regional Ethics Committee of Western Norway (REK Vest, approval number 2015/537). The trial was registered at ClinicalTrials.gov (identifier: NCT02721420). Before enrolment, written informed consent was obtained from the legal guardians of participating children.

## Data collection

All trial participants were followed for the full treatment period (10 weeks). The data for potential predictors were collected by trial personnel during caregiver interviews and medical assessments of participating children following their enrolment at the study hospital. Data were collected in the local language, Chichewa, and recorded in English using Open Data Kit software [16]. To assess adherence, the trial team collected blister packs at the participants' homes and performed tablet counts during unannounced, monthly visits following each course's 3-day administration period. The trial team was not blinded during this primary outcome assessment [16].

## Predictors

Potential predictors were considered along the three categories from the UNICEF Extended Model of Care: predictors focused on the child, predictors related to the caregiver and their behavior, and predictors pertaining to their household's resources [17, 18].

Child-related predictors included key demographic details of a child, such as sex and age, anthropometric measures, and hemoglobin level. In addition, a child's malaria-related medical history was considered, including the number of prior malaria infections and malaria-related hospital admissions. Predictors of caregiver behavior and resources included their demographic information, literacy, and educational status, religious affiliation, and tribe, as well as an inquiry on single parenting, number of children, and experience of child death. Some caregiving health behaviors were also included, such as whether a child slept under an insecticide-treated bed net (ITN). Predictive factors related to household resources included a household's socio-economic status (SES, in quintiles) based on an index of various assets, including

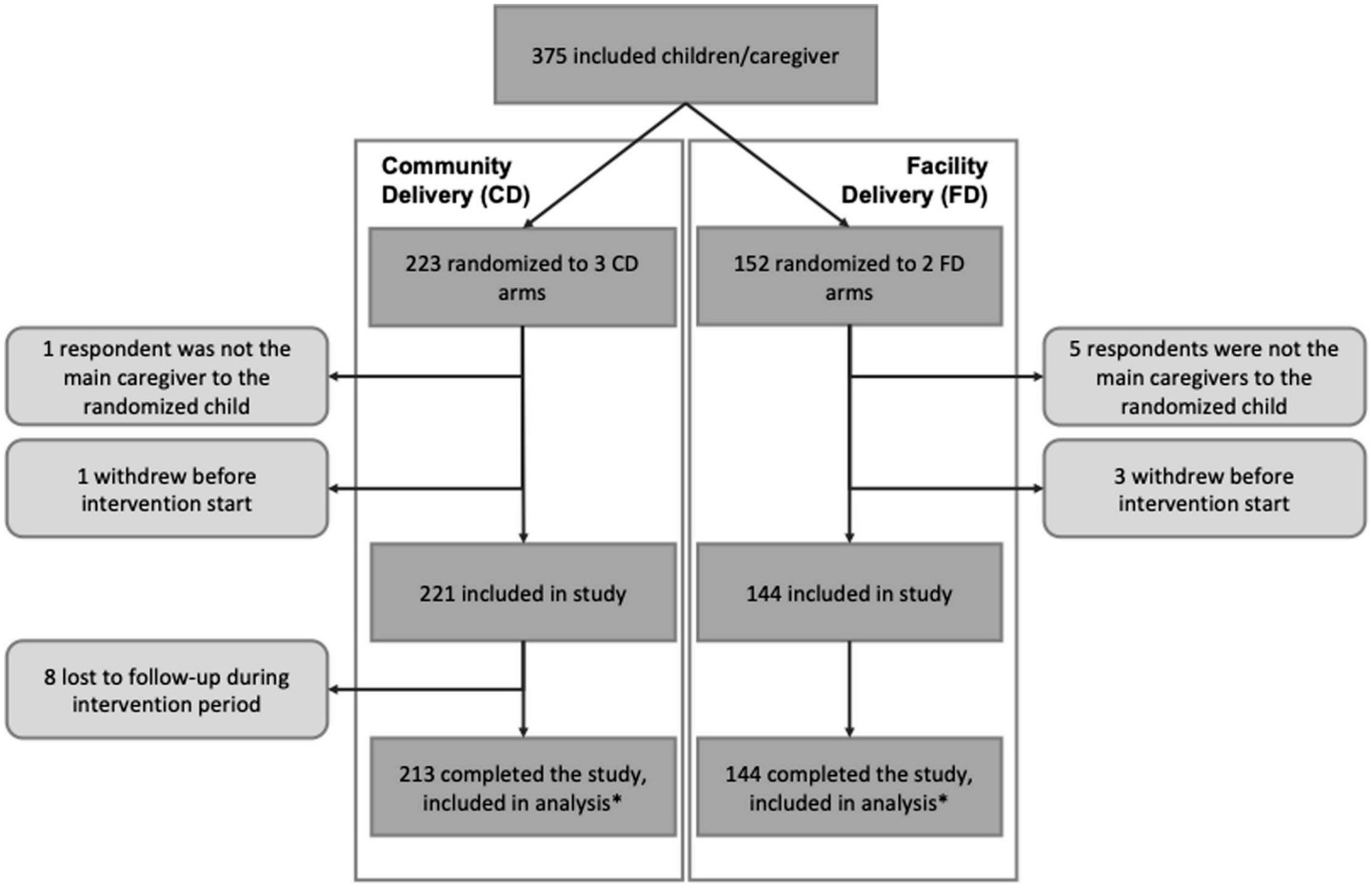

* Completion refers to data collection until intervention completion or death. Two children died in each intervention arm, however, provided data until they died. Participants that were lost to follow-up or died during the post-intervention period were not excluded.

**Fig 1. Study profile based on trial data from Gondwe et al, 2021 [8].**

household items, available resources, livestock possession, dwelling size, building materials, and sanitation facilities. We used principal component analysis to create this wealth index from these variables and multicollinearity analysis to adjust it further (S1 Text; S1 Table; S1 and S2 Figs). Household members owning their dwelling, being connected to the electricity grid, being able to rely on a regular income, and owning a bank account, were factors that we considered potentially important individual predictors for adherence. We therefore removed them from the index and tested them as separate predictors in this category. We also included community-related factors: the kind of drinking water source they used, its distance, and coverage of community-level malaria control efforts, particularly indoor residual spraying. Distance from households to the study hospitals was also considered.

All participants in the PDMC trial received the same preventive treatment either through the community-based or facility-based PDMC delivery mechanism. Adherence to PDMC was the primary outcome [8]. Due to this design, our analysis considered distribution strategy as its own category outside the three UNICEF categories.

## Statistical analysis

We expressed 'full adherence' as a binary outcome, defined as administering all nine DP doses over three months (i.e. three monthly DP courses consisting of three tablets each to be given on three consecutive days). Adherence was assessed by presenting three empty blister packs that contained three tablets each. Not returning all three blister packs empty at unannounced visits a few days after each course was termed 'non-adherence', irrespective of whether adherence was self-reported. Adherence data of caregivers whose children died during the trial was censored after the last course when the child was still alive to allow for 'full adherence' if the death occurred before they completed the three-course DP regimen (Fig 1).

Our analysis followed three steps. First, we tabulated each potential predictor by the adherence outcome. We present frequencies and percentages for categorical predictors and mean with standard deviations (SD) for continuous predictors. Thereafter, we conducted predictor analysis and report relative risks (RR) (95% confidence intervals) where adherence was the dependent variable, and each predictor was the independent variable [19]. We used a generalized linear model (GLM) for the Poisson family with a log link and robust variance estimation adjusting for clustering and study arm allocation [20]. The statistical significance of categorical variables was tested per subgroup category and for the entire variable using Wald testing. The Intra-Cluster Correlation coefficient (ICC) in the trial analysis was found to be insignificantly small (0.000008) [8]. This also applies to this secondary analysis, where 357 caregiver-child pairs came from 301 clusters.

Lastly, we included all statistically significant predictors at the p<0.05 level in a multivariable model [21, 22]. We tested for interaction with age and sex of both child and caregiver in the initial analysis. We also tested the crude and adjusted analyses for each treatment arm separately in view of the strong treatment effect. All variables included in the final model were tested for multicollinearity. Model performance was assessed by calculating the k-fold cross-validated area under the receiver operating characteristic (ROC)-curve with statistical inference obtained by bootstrapping [23].

Considering the wide distribution within the non-adherent group (zero to eight tablets taken) we created sub-categories, as defined in the previous PDMC trial and the cost-effectiveness analyses: *no or low* (zero to 2 tablets), *medium* (three to less than six tablets), and *high* (six to eight tablets) adherence (Fig 2) [8, 24]. We then conducted ordered logistic regression analysis for this categorical outcome, to test if this resulted in a different predictor selection. Accounting for the smaller sample size, we also inspected each potential predictor's (p-values <0.2) mean prevalence across these groups.

We used the Stata SE statistical analysis software package, version 17. We developed and reported this predictor model according to the EQUATOR TRIPOD-statement [25].

## Results

A total of 357 caregiver-child pairs were included in this analysis, of which 213 (60%) had been randomly allocated to community-based PDMC and 144 (40%) to facility-based PDMC (Fig 1). More males than females were enrolled in the trial. The z-scores (mean, SD) were: height-for-age (-1.67, 1.49), weight-for-age (-0.94, 1.06), and weight-for-height (-0.01, 1.17). The corresponding proportions of stunting, underweight, and wasting were 40%, 16% and 4%, respectively. Previous malaria infections were common; 61% had experienced at least one diagnosed malaria infection within the year before their hospital admission, and 9% at least four infections. Approximately four out of five children slept under ITNs. Most caregivers were mothers (94%), and the other caregivers were other family members. Their mean age was 29 years. Approximately one in four was a single parent, and one in five had previously experienced the

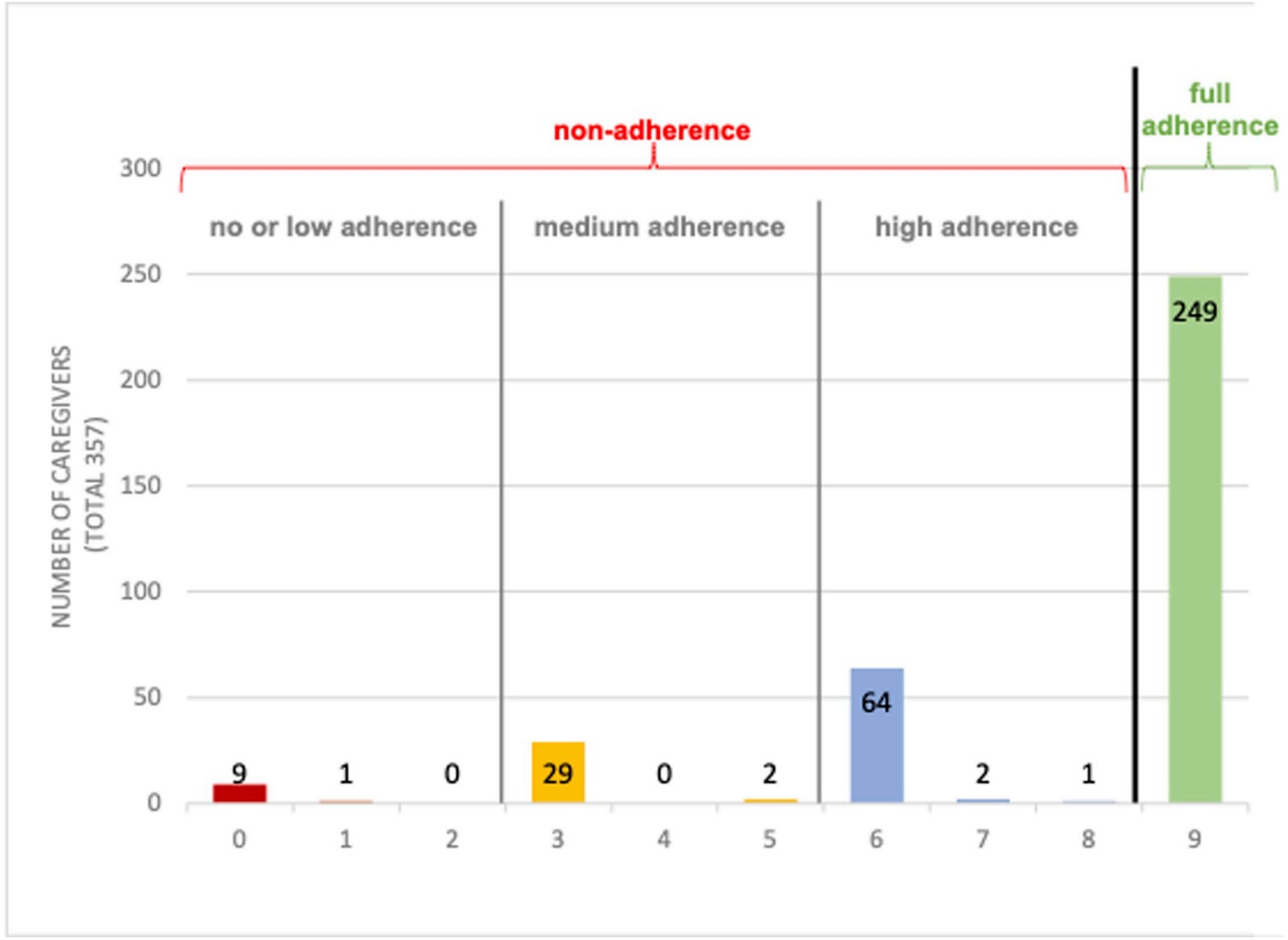

**Fig 2. Distribution of adherence behavior: The total number of tablets administered per caregiver.**

death of a child. Almost one in three caregivers was illiterate, and 14% had no education or had not completed primary school. Half of the caregivers had completed upper primary school. 98% of the households had no electricity. Less than 5% used surface water as the main source of drinking water (Table 1).

Out of the included 357 children/caregiver couples, 249 (70%) had full PDMC adherence, and 108 (30%) were categorized as "not adherent" (Table 1). The non-adherent category mostly received either zero, three, or six out of the nine tablets, reflecting that missing doses often involved skipping entire monthly course(s) of three tablets rather than one or two days of a 3-day course (Fig 2) [8]. Four included children died during the study period, all of whom were determined to be fully adherent.

As expected, the allocation to the trial's interventions showed a strong risk of non-adherence associated with facility-based PDMC, compared to community-based PDMC (RR, 95% CI: 0.64, 0.55 to 0.76). None of the potential predictors on child characteristics were associated with adherence except that multiple previous malaria infections (four or more) in the past year were associated with poorer adherence. Among the potential caregiver-related predictors, the

**Table 1. Descriptive statistics and regression analysis of potential predictors for adherence to PDMC.**

| Predictors | | Descriptive statistics by outcome frequencies (percentages)—unless row indicated otherwise | | Generalized linear model-analysis | |
|---|---|---|---|---|---|
| Predictor categories Included potential predictor variables | Variable categories | Non-adherence n = 108 | Full adherence n = 249 | Crude relative risk (95% CI) | Adjusted relative risk (95% CI) |
| **Intervention allocation, PMC trial (Gondwe, 2021)** | | | | | |
| PMC delivery | community-based | 40 (37.0) | 173 (69.5) | 1 | 1 |
| | facility-based | 68 (63.0) | 76 (30.5) | 0.65 (0.55, 0.76)** | 0.64 (0.55, 0.76)** |
| **Characteristics of child at enrolment** | | | | | |
| Sex | male | 60 (55.6) | 144 (57.8) | 1 | |
| | female | 48 (44.4) | 105 (42.2) | 1.00 (0.87, 1.14) | |
| Child age in months (mean, SD)* | | 27.35 (13.18) | 30.14 (13.63) | 1.00 (0.99, 1.00) | |
| Child was stunted (Z<-2) | yes | 52 (48.2) | 92 (37.0) | 0.88 (0.76, 1.02) | |
| Child was wasted (Z<-2) | yes | 4 (3.7) | 11 (4.4) | 1.11 (0.85, 1.45) | |
| | missing | 1 (0.9) | 0.00 | | |
| Child was underweight (Z<-2) | yes | 3 (2.8) | 10 (4.0) | 0.91 (0.71, 1.17) | |
| | missing | 0.00 | 1 (0.3) | | |
| Height-for-age z-score (mean, SD)* | | -1.85 (1.15) | -1.60 (1.60) | 1.02 (0.99, 1.06) | |
| Weight-for-height z-score (mean, SD)* | | -0.11 (1.05) | 0.04 (1.22) | 1.03 (0.98, 1.09) | |
| Weight-for-age z-score (mean, SD)* | | -1.11 (1.07) | -0.87 (1.05) | 1.06 (0.99, 1.12) | |
| Hemoglobin level in g/dl (mean, SD)* | | 8.11 (1.57) | 7.91 (1.41) | 0.98 (0.93, 1.02) | |
| Four or more malaria infections, past year | yes | 5 (4.6) | 28 (11.2) | 0.82 (0.70, 0.96)** | 0.83 (0.71, 0.97)** |
| Hospital admission for malaria, past year | no | 100 (92.6) | 223 (89.6) | 0.84 (0.37, 1.90) | |
| Child slept under mosquito net during the past night | no | 21 (19.4) | 58 (23.3) | 0.96 (0.80, 1.15) | |
| Four or more siblings | yes | 19 (17.6) | 59 (23.7) | 0.93 (0.80, 107) | |
| **Characteristics of caregiver and caregiving behavior at enrolment** | | | | | |
| Caregiver is the mother | no | 7 (6.5) | 15 (6.0) | 0.92 (0.71, 1.20) | |
| Caregiver's age in years (mean, SD)* | | 28.8 (7.79) | 29.35 (8.66) | 1.01 (0.99, 1.01) | |
| Caregiver is a single parent | yes | 31 (28.7) | 63 (25.3) | 0.94 (0.80, 1.09) | |
| Caregiver experienced previous child death | yes | 21 (19.4) | 46 (18.5) | 1.01 (0.85, 1.20) | |
| Caregiver is illiterate | yes | 30 (27.8) | 81 (32.5) | 0.95 (0.83, 1.09) | |
| Caregiver's highest completed education level*** | none | 10 (9.3) | 39 (15.7) | 1 | 1 |
| | lower primary | 30 (27.8) | 55 (22.9) | 0.80 (0.66, 0.98)** | 0.78 (0.64, 0.95)** |
| | upper primary | 59 (54.6) | 120 (48.2) | 0.83 (0.70, 0.97)** | 0.79 (0.67, 0.92)** |
| | lower secondary, higher | 9 (8.3) | 35 (14.1) | 1.01 (0.84, 1.22) | 0.98 (0.80, 1.21) |
| Caregiver's religion | Christian | 86 (79.6) | 187 (75.1) | 1 | |
| | other | 22 (20.4) | 62 (24.9) | 0.99 (0.86, 1.15) | |
| Caregiver's tribe | Chewa | 13 (12.0) | 40 (16.1) | 1 | |
| | Yao | 35 (32.4) | 77 (30.9) | 0.87 (0.73, 1.04) | |
| | Lomwe | 30 (27.8) | 80 (32.1) | 0.94 (0.79, 1.11) | |
| | Nyanja | 22 (20.4) | 35 (14.1) | 0.83 (0.64, 1.06) | |
| | others | 8 (7.4) | 17 (6.8) | 0.91 (0.68, 1.23) | |
| Caregiver has experience giving medicine to this child | no | 12 (11.1) | 24 (9.6) | 0.97 (0.76, 1.23) | |
| **Household's caregiving resources** | | | | | |

*(Continued)*

**Table 1.** (Continued)

| Predictors | | Descriptive statistics by outcome frequencies (percentages)—unless row indicated otherwise | | Generalized linear model-analysis | |
|---|---|---|---|---|---|
| Number of adults in household (mean, SD)* | | 2.06 (0.85) | 2.21 (0.94) | 1.05 (0.98, 1.12) | |
| Caregiver could report a source of main income | no | 34 (31.5) | 76 (30.5) | 0.98 (0.85, 1.13) | |
| Distribution by socioeconomic index in quintiles*** | poorest quintile | 31 (28.7) | 59 (23.7) | 1 | 1 |
| | 2nd quintile | 10 (9.3) | 50 (20.1) | 1.20 (1.01, 1.42)** | 1.23 (1.04, 1.42)** |
| | 3rd quintile | 32 (29.6) | 42 (16.9) | 0.83 (0.66, 1.05) | 0.80 (0.64, 1.01) |
| | 4th quintile | 18 (16.7) | 49 (19.7) | 1.06 (0.87, 1.29) | 1.04 (0.85, 1.26) |
| | richest quintile | 17 (15.7) | 49 (19.7) | 1.15 (0.95, 1.39) | 1.09 (0.89, 1.32) |
| Household member owns residential home | no | 16 (14.8) | 23 (9.2) | 0.81 (0.63, 1.05) | |
| At least one Household member has a bank account | no | 102 (94.4) | 232 (93.2) | 1.06 (0.84, 1.34) | |
| | do not know | 0 (0) | 2 (0.8) | | |
| Household has electricity | no | 107 (99.1) | 243 (97.6) | 0.80 (0.54, 1.18) | |
| Travel distance to clinic, straight line, in km (mean, SD)* | | 19.83 (8.89) | 19.64 (9.10) | 0.99 (0.99, 1.01) | |
| Household has water access within 10 min walk | no | 44 (40.7) | 113 (45.4) | 1.05 (0.92, 1.20) | |
| Source of drinking water used by the household | piped water (improved) | 17 (15.7) | 38 (15.3) | 1 | |
| | pumped ground water (improved and non-improved) | 82 (75.9) | 204 (81.9) | 1.03 (0.86, 1.24) | |
| | surface water (non-improved) | 9 (8.3) | 7 (2.8) | 0.66 (0.36, 1.20) | |

* Descriptive statistics for continuous variables were calculated using t-test.

** Predictors with p-values <0.05.

*** Multilevel variables that were significant as entire variable (p<0.05), calculated using Wald-test.

caregivers' education showed a significant association with adherence behavior. However, high adherence was correlated with 'no or no completed education'. Compared to this group, having completed lower or upper primary education was associated with higher non-adherence (RR, 95% CI: 0.78, 0.64 to 0.95; and 0.79, 0.67 to 0.92, respectively). At the household level, the socio-economic index showed a mixed picture, where the middle group adhered most poorly.

The model's performance, adjusted for k-fold cross-validation, was acceptable, with the mean area under the ROC-curve estimated to be 0.65 (95%CI: 0.57 to 0.71). The analysis with the non-adherent group separated into non-adherent sub-categories (*high but not full*, *medium*, and *low or no* adherence) did not yield significant predictors, we do not report this analysis. We neither found any important differences comparing the mean occurrence of potential predictors in these sub-groups, each compared among each other and to the fully adherent group.

## Discussion

We developed a prognostic multivariable model to analyze determinants of adherence of Malawian caregivers to PDMC, the first predictor analysis for adherence to PDMC. Our results

are mixed, and we cannot explain all findings, although we included key predictors for caregiver adherence as established in the literature in comparable contexts. Some uncertainty remained in measuring the adherence-outcome as a few caregivers were repeatedly not home during control visits, while few others self-reported adherence but having lost or thrown away the empty blister pack. Such problems are recognized in the research on ACT adherence; however, the alternative of self-reporting has been shown to deviate markedly from actual adherence [11, 12].

Two systematic reviews from 2014 of ACT adherence summarized predicting factors for non-adherence to curative malaria treatment with ACTs, i.e., not for prevention. Both reviews reported caregivers' adherence separately from adults' adherence, when minor patients were included [11, 12]. Relatively older caregivers were generally associated with higher adherence levels to ACTs, an association we cannot confirm in our PDMC study. Likewise, higher education levels of caregivers were reported to correlate with improved adherence to ACTs. Our findings suggest an opposite correlation where no completed education, the lowest category, was associated with significantly higher adherence than the next two higher categories (completed lower and upper primary school, respectively). This result may be related to the trial setting where particular attention was given to illiterate caregivers' information and consent procedures, during enrolment, and when instructing them in drug administration. We cannot determine if this has affected our population, but others have demonstrated that a good patient-provider relationship is among the most consistent predictors for improved adherence [26]. Speaking the language of administration instructions, or demonstrably understanding these instructions, was likewise associated with higher adherence in the literature on ACT adherence. The trial offered instructions in Chichewa, the most used language in Southern Malawi, widely spoken in all households.

Relatively low income or socio-economic status has been associated with poor adherence behavior [11, 12]. Contradicting this association, our SES-index indicates mixed directions of adherence behavior across the quintiles. This index, however adjusted, generated a skewed distribution, displaying relatively small differences among the households in the four poorer quintiles. It is possible that our asset-based data included in the index were not sufficiently sensitive to separate this rural population into more substantially different quintiles. Skewedness is a recurrent challenge of asset-based indices in comparable socio-economic settings [27].

Caregivers may well adhere differently to a regimen depending on whether they are treating a notably sick child that shows a positive cause-effect response to their caring, or giving the same regimen as prophylaxis to a seemingly healthy child, without such causal learning [28]. Instead, the direct effect of a preventive regimen may more likely be perceived as "neutral", or even "negative" in case of side-effects like occasional vomiting in case of PDMC-DP [29, 30]. The generally established complexity behind the drivers to adhere to curative treatments may be even greater in case of preventive treatments, especially for caregiver-child relationships. The perceived severity of a child's disease, for example, has been reported as a predictor for increased adherence, specifically for ACT treatment [11, 31]. This determinant cannot be directly translated to PDMC, where a future severity is uncertain and more abstract. Experiencing repeated non-severe malaria infections in a child was associated with poorer caregiver adherence to PDMC.

Prior malaria-related hospital admissions of a child indicate a caregiver's experience of caring for a severely sick child. These experiences from the past may have stimulated caregivers' adherence to PDMC in the same direction as perceived severity increases adherence to curative treatment; however, we did not find this association for PDMC.

Due to the small sample size, we cannot rule out type II errors (not distinguishing a true negative finding from non-identification). In addition, while the data collected was

comprehensive and structured along the framework we used, a more targeted inquiry towards caring attitudes and parenting behavior may have offered a deeper understanding of the decisive actors' motivations and capacities: the caregivers. Understanding their behavior and capacities remains important to tailor implementation mechanisms and patient communication towards improved adherence to PDMC in its given complexity. Future implementation research may thus consider pooling or collecting a larger data sample to better address this. Additionally, qualitative inquiry on regimen experience and adherence motivators may help clarify some of our mixed results. Finally, as we reveal no obvious amendable determinants for poor adherence that can be considered during the roll-out of PDMC programs, implementation efforts need to ensure high general fidelity to programs to achieve high adherence rates among caregivers.

## Conclusion

We investigated potential determinants for PDMC adherence of rural caregivers in Malawi and we found no implementation-relevant predictor for their adherence behavior. Our results are mixed and in disagreement with the literature on adherence to ACT treatment in children. It is possible that, compared to malaria treatment, malaria prevention introduces more complexity in caregivers' adherence behavior due to, for example, the absence of an illness to be treated. The analyses reveal no obvious determinants for poor adherence that can be targeted and instead PDMC-programs needs to maximize implementation fidelity to achieve high adherence.

## Supporting information

**S1 Text. Summary of methods and results behind the index-variable for households' socio-economic status (SES).**
(DOCX)

**S1 Table. Overview of variables considered in the predictor analysis.**
(DOCX)

**S1 Fig. The eigenvalues for the 11 principal components included in the adjusted analysis.**
(DOCX)

**S2 Fig. Households' relative socio-economic status based on adjusted PCA-analysis, separated into quintiles.** PCA: Principal Component Analysis.
(DOCX)

## Acknowledgments

We acknowledge the Training and Research Unit of Excellence (TRUE) for the data collection and logistical support in the PDMC trial in Malawi. We are thankful for the support we received from the pediatric department at Zomba Central Hospital and the Zomba District Health Office. We thank all the caregivers and their children who participated in the trial. Lastly, we thank our colleague Peter Hangoma for his input when we revised our analysis.

## Author Contributions

**Conceptualization:** Melf-Jakob Kühl, Feiko O ter Kuile, Bjarne Robberstad.

**Data curation:** Melf-Jakob Kühl, Thandile Nkosi-Gondwe, Kamija S Phiri, Mehmajeet Pannu, Mavuto Mukaka.

**Formal analysis:** Melf-Jakob Kühl, Mavuto Mukaka, Ingunn M. S Engebretsen.

**Funding acquisition:** Kamija S Phiri, Bjarne Robberstad.

**Investigation:** Melf-Jakob Kühl, Feiko O ter Kuile, Kamija S Phiri, Bjarne Robberstad.

**Methodology:** Melf-Jakob Kühl, Mehmajeet Pannu, Bjarne Robberstad, Ingunn M. S Engebretsen.

**Project administration:** Thandile Nkosi-Gondwe, Kamija S Phiri, Bjarne Robberstad.

**Supervision:** Bjarne Robberstad, Ingunn M. S Engebretsen.

**Validation:** Mavuto Mukaka, Ingunn M. S Engebretsen.

**Visualization:** Melf-Jakob Kühl, Ingunn M. S Engebretsen.

**Writing – original draft:** Melf-Jakob Kühl, Ingunn M. S Engebretsen.

**Writing – review & editing:** Melf-Jakob Kühl, Thandile Nkosi-Gondwe, Feiko O ter Kuile, Kamija S Phiri, Mehmajeet Pannu, Mavuto Mukaka, Bjarne Robberstad, Ingunn M. S Engebretsen.

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
