## [Decision Letter · Decision Letter 0]

7 Oct 2022

PGPH-D-22-01351

Predicting caregivers’ adherence to postdischarge malaria chemoprevention in Malawian pre-school children: a prognostic multivariable analysis

Dear Dr. Kühl,

Thank you for submitting your manuscript to PLOS Global Public Health. After careful consideration, we feel that it has merit but does not fully meet PLOS Global Public Health’s publication criteria as it currently stands. Therefore, we invite you to submit a revised version of the manuscript that addresses the points raised during the review process.

We look forward to receiving your revised manuscript.

Kind regards,

Ruth Ashton, Ph.D.

Academic Editor

Journal Requirements:

1. Please provide separate figure files in .tif or .eps format only and remove any figures embedded in your manuscript file. Please also ensure that all files are under our size limit of 10MB.

Additional Editor Comments (if provided):

Reviewers' comments:

Reviewer's Responses to Questions

**Comments to the Author**

1. Does this manuscript meet PLOS Global Public Health’s publication criteria? Is the manuscript technically sound, and do the data support the conclusions? The manuscript must describe methodologically and ethically rigorous research with conclusions that are appropriately drawn based on the data presented.

Reviewer #1: Yes

Reviewer #2: Yes

Reviewer #3: Yes

2. Has the statistical analysis been performed appropriately and rigorously?

Reviewer #1: No

Reviewer #2: Yes

Reviewer #3: Yes

3. Have the authors made all data underlying the findings in their manuscript fully available (please refer to the Data Availability Statement at the start of the manuscript PDF file)?

Reviewer #1: Yes

Reviewer #2: Yes

Reviewer #3: Yes

4. Is the manuscript presented in an intelligible fashion and written in standard English?

Reviewer #1: Yes

Reviewer #2: Yes

Reviewer #3: Yes

5. Review Comments to the Author

Reviewer #1: Introduction

The authors indicated the need for this study and included very relevant and recent publication in this area. PDMC has been shown to reduce post-discharge mortality and hospital readmission among children with malaria/anaemia and therefore improving its adherence should be encouraged by all means.

Methods

-The authors need to add more details about the current study. I think they should make clear this was a secondary analysis of previous cluster randomized trial (page 4 lines 9/17).

-The sample size was powered for original trial end point. Do you think it was adequate to answer the new question on adherence? Maybe a post-hoc power could have helped detect if the study was adequately powered.

-The original design was a cluster randomized trial, why was the initial design ignored in the prognostic model. I think not accounting for the clustering in the original trial was grave mistake in this modelling.

-Did the authors collect any new data for this study or all data were collected during the original trial (Page 6 lines 3/12). If data on adherence was collected after study completion, it could be subject to reporting bias. Some caregivers could have lost the blister packs.

-The authors have listed the predictors explored in this study and report the predictors were based on UNICEF Extended Model of Care’s three categories. The model creates a hierarchy of variables from the enabling predictors to underlying and lastly the immediate predictors. The implication of this model is that there could be many interactions i.e some enabling predictors could directly impact adherence or indirectly through underlying or immediate predictors. The same would be true for underlying predictors. Using a `flat’ model like logistic regression may not adequate test these interactions. Structural equation modelling would have adequately addressed the interactions and different paths.

-Which variables and what method was used to create the household wealth index?

-Another predictor that probably would impact adherence is access to health care given one arm was receiving their drugs at the clinic. The authors collected data on travel time to the study hospital but this predictor was not included in the model.

-Was data on maternal depression available?

Results

- I think it would be challenging to interpret the effect of household wealth index on adherence since the authors have not reported the asserts included in the score and the method of computing the score.

-Approximately 6% caregivers were not biological caregivers, who were they? Would it make sense to run sensitivity analysis with only biological caregivers?

-I suspect the model could not work because there were many highly correlated variables for example were all the child anthropometry like HAZ, WAZ, WHZ, stunting, wasting, underweight etc included in the model simultaneous?

-What was the discriminatory value of the multivariable model? How did the model perform? Did you calculate a measure like AUCs?

-Looking at the model and predictors included in the model, is it correct to say thi is a caregiver predictors model? Both child and care giver predictors are included.

Discussion

-Although the authors report their findings are inconclusive, I think they did not put serious thought in modelling including predictors selection, selection of regression model etc.

-Logistic regression overestimates risk ratios when the prevalence >10%. Literature suggests alternative models to estimate relative risk rather than odds ratio. Here are some useful literature: https://www.ncbi.nlm.nih.gov/pmc/articles/PMC3348192/

https://www.ncbi.nlm.nih.gov/pmc/articles/PMC2292207/

https://academic.oup.com/aje/article/157/10/940/290159

-Alternative the first step in this case could have been to start with qualitative work about adherence before fixing prognostic model.

Reviewer #2: A good paper on the predictors of caregivers adherence with a good comprehension of the immediate, underlying and enabling predictor. However, the paper could be improved in the following ways;

1. In your methods you mentioned that the variables with p-values <0.2 would be included in the adjusted logistic regression, with the results indicated on Table 1. However, it is noted that some logistic results that met the criteria were not included in the adjusted results and some results which did not meet the criteria had adjusted logistic results. Either this was a minor error or oversight with the superscript labelling, or the criteria wasn’t followed strictly. Otherwise, please clarify why the criteria was not strictly followed.

2. Most of the predictors were not significant as stand-alone variables. It would be better if an interaction term was explored between the categories of predictors. In a real-life situation, these predictors all interact to influence the behaviour of a caregiver. This could yield more significant results and it would be interesting to see how these predictors interact.

3. Please clarify the method used to calculate the socio-economic variable; was this done using factor analysis (principal component analysis)? What did you use to inform your asset index questionnaire? Was it the DHS? Or the country’s establish wealth index? Also, you tried to differentiate between the access of water variable in the SES index and the community source of water, which is confusing as the community source of water and household access to that source of water, all make up the household’s access to water. This makes the community source of water variable redundant and hence insignificant as it is already accounted for in the SES index results. This also goes for the household’s access to electricity which is also part of the SES index. Suggest you either remove community source of water and household electricity, as separate variables, from the logistic regression or give a clearer explanation on how these variables differ from the SES index as predictors.

Reviewer #3: My main suggestion is on the choice of outcome and specifically the decision to categorise and report adherence as a binary 'all or nothing' (p8, line 6-8) rather than considering one (or more) category of partial adherence. The mean number of tablets taken by the non-adherent group (x̄=4.7, SD=2, range=0 to 8) shows the vast majority of those did take at least some tablets, but that there was substantial variation, including those never took and those who took almost all.

While partial adherence is of course sub-optimal, I think doing this would make the analysis potentially more informative in identifying risk factors for poor adherence more clearly than it does currently, which the authors concede is of limited utility. It may (or may not!) be the case that by grouping many different sets of adherence patterns together some risk factors are being obscured.

Connected to this – I would strongly suggest that even if the authors chose not to take the analysis in this direction, that the patterns of adherence are reported in more detail. This could be very valuable from a programmatic perspective, even if no clear risk factors are identified. What exactly is going on in the ‘not full adherence’ group – is it that one dose is being taken in full, the second in part, then the third not at all; or is it one to two tablets of each dose? Or is it just a very mixed picture with no clear pattern. Even descriptive analysis would be valuable given PDMC is quite novel and help contextualise the analysis.

Other comments:

Page 2, line 12 – why are these associations necessarily counter intuitive (and to who)? This is repeated (page 14). In the discussion section the authors even go on to speculate that those with lower levels of literacy may receive better counselling from healthcare workers and cite some evidence to support this. I think it would be more appropriate to describe this without value judgement (e.g. just the direction of association, is it positive or negative).

Page 3, line 1 – specifically, WHO recommended PDMC.

Page 5, line 3 –should this read “their” as later on (page 9, line 10) it is specified that not all caregivers were mothers.

Page 7, line 18 – please clarify how the “asset-based index” was calculated (PCA?) and what specific covariates were included (“several household features” is not clear).

Page 8, line 11 – why would observed treatment (I presume this was a small number though) not be considered reliable way of measuring adherence?

Page 13, line 3-5 – can you map these to standard categorisation of water source (e.g. how do (protected) dug well or spring fit into these)?

Page 14, line 23-25 – would the authors consider reporting adherence reported by self report and/or direct observation as a supplementary table?

Page 14-17 – Would the authors consider citing evidence about adherence to treatment from outside malaria that could be relevant to this discussion more broadly? For example, I would be surprised if there was not evidence from community use of antibiotics etc (see Non-adherence to oral antibiotics for community paediatric pneumonia treatment in Malawi – A qualitative investigation).

Page 15, line 13 – I am not sure this sentence is necessary, as the authors go on to give a reasonable explanation based on the literature.

Page 15, line 21-22 – I am not sure the word “mastered” is appropriate in this context, perhaps “widely spoken”.

Page 16, line 3 – again, I am not sure the word “inexplicable” is the best choice here, “mixed” is sufficient.

6. PLOS authors have the option to publish the peer review history of their article (what does this mean?). If published, this will include your full peer review and any attached files.

**Do you want your identity to be public for this peer review?** For information about this choice, including consent withdrawal, please see our Privacy Policy.

Reviewer #1: **Yes: **Dr Moses Ngari

Reviewer #2: No

Reviewer #3: No

---

## [Decision Letter · Decision Letter 1]

1 Mar 2023

PGPH-D-22-01351R1

Predicting caregivers’ adherence to postdischarge malaria chemoprevention in Malawian pre-school children: a prognostic multivariable analysis

Dear Dr. Kühl,

Thank you for submitting your manuscript to PLOS Global Public Health. After careful consideration, we feel that it has merit but does not fully meet PLOS Global Public Health’s publication criteria as it currently stands. Therefore, we invite you to submit a revised version of the manuscript that addresses the points raised during the review process.

The reviewer has acknowledged that all previous comments have been addressed, however they have flagged a few minor items for your consideration and response. 

We look forward to receiving your revised manuscript.

Kind regards,

Ruth Ashton, Ph.D.

Academic Editor

Journal Requirements:

Additional Editor Comments (if provided):

Reviewers' comments:

Reviewer's Responses to Questions

**Comments to the Author**

1. If the authors have adequately addressed your comments raised in a previous round of review and you feel that this manuscript is now acceptable for publication, you may indicate that here to bypass the “Comments to the Author” section, enter your conflict of interest statement in the “Confidential to Editor” section, and submit your "Accept" recommendation.

Reviewer #1: All comments have been addressed

2. Does this manuscript meet PLOS Global Public Health’s publication criteria? Is the manuscript technically sound, and do the data support the conclusions? The manuscript must describe methodologically and ethically rigorous research with conclusions that are appropriately drawn based on the data presented.

Reviewer #1: Yes

3. Has the statistical analysis been performed appropriately and rigorously?

Reviewer #1: Yes

4. Have the authors made all data underlying the findings in their manuscript fully available (please refer to the Data Availability Statement at the start of the manuscript PDF file)?

Reviewer #1: Yes

5. Is the manuscript presented in an intelligible fashion and written in standard English?

Reviewer #1: Yes

6. Review Comments to the Author

Reviewer #1: Thank for addressing the comments, the manuscript reads much better now. However, I have one major and minor comments.

Major comment

I am still struggling with the title of the study. I would remove the word 'caregiver' from the title so that it reads `Predicting adherence to PDMC ....'. It is because we are talking about PDMC on children and not caregivers.

Minor comments

a) Page 11, line 7 on the results, the author report the measure of effect PR rather than RR (PR, 95% 0.65, 0.55 to 0.76).

b) Page 12, line 7, are the measures of effect OR or RR?

c) Can you add 95%CI to the AUC? You could use bootstrapped AUC (say with 1000 replications for internal validation).

7. PLOS authors have the option to publish the peer review history of their article (what does this mean?). If published, this will include your full peer review and any attached files.

**Do you want your identity to be public for this peer review?** For information about this choice, including consent withdrawal, please see our Privacy Policy.

Reviewer #1: **Yes: **Dr Moses Ngari

---

## [Editor Report · Decision Letter 2]

14 Mar 2023

Predicting adherence to postdischarge malaria chemoprevention in Malawian pre-school children: a prognostic multivariable analysis

PGPH-D-22-01351R2

Dear Mr Kühl,

We are pleased to inform you that your manuscript 'Predicting adherence to postdischarge malaria chemoprevention in Malawian pre-school children: a prognostic multivariable analysis' has been provisionally accepted for publication in PLOS Global Public Health.

Best regards,

Ruth Ashton, Ph.D.

Academic Editor